# Simple and Equipment-Free Paper-Based Device for Determination of Mercury in Contaminated Soil

**DOI:** 10.3390/molecules26072004

**Published:** 2021-04-01

**Authors:** Hikmanita Lisan Nashukha, Jirayu Sitanurak, Hermin Sulistyarti, Duangjai Nacapricha, Kanchana Uraisin

**Affiliations:** 1Flow Innovation-Research for Science and Technology Laboratories (Firstlabs), Mahidol University, Rama 6 Road, Bangkok 10400, Thailand; hikmanitalisan@gmail.com (H.L.N.); jsitanurak@gmail.com (J.S.); dnacapricha@gmail.com (D.N.); 2Department of Chemistry and Center of Excellence for Innovation in Chemistry, Faculty of Science, Mahidol University, Rama 6 Road, Bangkok 10400, Thailand; 3Department of Chemistry, Faculty of Science, Brawijaya University, Jl. Veteran Malang 65145, Indonesia; sulistyarti@yahoo.com

**Keywords:** tetraiodomercurate, mercury, paper-based, iodometry, soil, water

## Abstract

This work presents a simple and innovative protocol employing a microfluidic paper-based analytical device (µPAD) for equipment-free determination of mercury. In this method, mercury (II) forms an ionic-association complex of tetraiodomercurate (II) ion (HgI_4_^2−^_(aq)_) using a known excess amount of iodide. The residual iodide flows by capillary action into a second region of the paper where it is converted to iodine by pre-deposited iodate to liberate I_2(g)_ under acidic condition. Iodine vapor diffuses across the spacer region of the µPAD to form a purple colored of tri-iodide starch complex in a detection zone located in a separate layer of the µPAD. The digital image of the complex is analyzed using ImageJ software. The method has a linear calibration range of 50–350 mg L^−1^ Hg with the detection limit of 20 mg L^−1^. The method was successfully applied to the determination of mercury in contaminated soil and water samples which the results agreed well with the ICP-MS method. Three soil samples were highly contaminated with mercury above the acceptable WHO limits (0.05 mg kg^−1^). To the best of our knowledge, this is the first colorimetric µPAD method that is applicable for soil samples including mercury contaminated soils from gold mining areas.

## 1. Introduction

Mercury is well known as one of the most toxic elements for organisms and human health. It is known that natural disasters such as volcano eruptions and forest fires can cause the release of mercury and contamination of the environment [1]. Nonetheless, emission of mercury from artisanal small-scale gold mining (ASGM) is the largest source of mercury emission in some developing countries [2,3], where mercury is used for amalgamation and purification of gold [4]. The US EPA methods for determination of mercury in water [5] and soil [6] are based on cold vapor atomic absorption spectrometry (cold vapor-AAS). For complicated matrices such as soil and sediment, a method employing use of an alkaline reagent (pH ≈ 14), named Universol^®^ was recently presented in association with cold vapor-AAS [7]. There are also other equipment-based methods that have been employed for these samples, such as high-performance liquid chromatographic method using chemiluminescence detection [8] and inductively coupled plasma mass spectrometric method or ICP-MS [9] for soil samples, as well as a resonance scattering spectroscopic method [10] for water samples. 

Among equipment-based techniques, colorimetric spectrometry is still the favored technique for the determination of mercury because of its simplicity and availability of cost-effective instruments. The water soluble Michler’s thioketone reagent (4.4′-Bis-(dimethylamino)-thiobenzophenone) has been used for detection of mercury but the method could be interfered by other metal ions, such as Cr(III), Fe(III) and Cu(II) [11]. Several selective reagents have also been proposed for the determination of mercury. However, most chromogenic reagents are insoluble in water, e.g., 1-[(5-Benzyl-1,3-thiazol-2-yl)diazenyl]naphthalene-2-ol [12], dithizone [13], 2-(3-hydroxy-1-methylbut-2-enylideneamino)pyridine-3-ol [14] and 2,4-bis(4-phenylazophenylaminodiazo) benzenesulfonic acid [15]. In order to improve the solubility of these organic reagents, the reaction has to be carried out in a micellar system of sodium dodecyl sulphate [13] or Triton X-100 [15]. Functionalized gold nanoparticles (AuNPs) with dithioerythritol [16], 3, 5-dimethyl-1-thiocarboxamidepyrazole (Pzl) [17] or mercaptophenyl boronic acid (MPBA) [18] have also been presented for colorimetric/spectrometric detection of mercury. Nonetheless, for low- and middle-income countries (in Asia, Africa, the Pacific and South America) where emissions of mercury are mainly from gold mining activities, equipment-free and low-cost devices for mercury determination are needed as tools for monitoring the anthropogenic emission at point source and the extent of the spread of the contamination. 

Microfluidic paper-based analytical devices (µPADs) [19,20,21,22,23], as well as reagent impregnated-paper strips/devices, are analytical tools that is in line with the strategy of equipment-free analysis using low-cost devices. For producing µPADs, patterns are drawn on the paper substrate using various hydrophobic materials to act as barriers to control the fluid flow on the paper. For mercury, there are some interesting paper-based devices that have been presented for colorimetric detection of mercury (as Hg^2+^) with photographic image analysis [24,25,26,27,28]. Patidar et al. presented two synthesized rhodamine derivatives for colorimetric detection of Hg^2+^ (and Cr^3+^) on paper strips and cellulose acetate membrane [24]. A resorufin thionocarbonate signaling probe for Hg^2+^ was recently synthesized by the group of Chang [25]. Hg^2+^ induces cleavage reaction of thionocarbonate moiety of the probe leading to a prominent color change from yellow to pink. This probe was later incorporated into a wax printed µPAD for selective detection of Hg^2+^. Nanoparticles have been utilized by various groups to develop sensitive colorimetric paper-based analytical devices (PADs)/µPADs for quantifying Hg^2+^ [26,27,28,29,30]. Han et al. presented a paper chip for Hg^2+^ detection based on the enzyme-like catalytic activity of gold nanoparticles (AuNPs) that is enhanced by the formation of Au-Hg amalgam for the reaction between 3,3′,5,5′-tetramethylbenzidine (TMB) and H_2_O_2_ [29]. The color intensity of the chromogenic peroxidase substrate (TMB image) corresponds to the concentration of Hg^2+^. Silver nanoparticles (AgNPs) were also employed as colorimetric sensors for PADs/µPADs [26,27,30]. Hg^2+^ is reduced by AgNPs which leads to disintegration of the AgNPs into smaller particles and formation of Hg^0^ [26]. Subsequent deposition of Hg^0^ onto AgNPs gives Ag-Hg amalgam particles resulting in the change in the color intensity which is dependent on the concentration to Hg^2+^ [30]. Pourresza et al. presented a colorimetric paper-based analytical device incorporating synthesized curcumin nanoparticles (CcNPs) for quantifying Hg^2+^ [28]. Complex formation between Hg^2+^ and CcNPs leads to the fading of the yellow color which is used for the determination of Hg^2+^. A distance-based readout µPAD for determination of Hg^2+^ was also presented by Cai et al. [31]. The reaction between Hg^2+^ and dithizone in NaOH solution forms an insoluble colored complex in the paper channel. The length of the pink reaction band increases linearly with the concentration of Hg^2+^. Most of these PAD/µPAD methods were tested to analyze water samples spiked with mercury [25,26,27,28,29,30,31]. To the best of our knowledge, there has been no PAD/µPAD previously presented for detection of mercury in soil. 

Therefore, this work presents a µPAD as tool for analyzing soil heavily contaminated with mercury. The µPAD can be used to identify the source and the dispersion of mercury emission from gold mining activities. The µPAD method is also appropriate for assessing inactive gold mines for persistence of the release of mercury to villages surrounding the mines. The µPAD was fabricated based on the concept of membraneless gas-separation microfluidic paper-based analytical device (membraneless gas separation µPAD) that was introduced in 2016 by Phansi et al. [32]. This work quantitates mercury via formation of tetraiodomercurate(II) ion (HgI_4_^2^^−^) with a known amount of excess iodide. This ionic-association complex is formed in the sample reservoir of the “donor layer”. The residual iodide reagent then flows via capillary action to react with iodate and acid to generate iodine (I_2_) in the neighboring reservoir of the “donor layer”. I_2_ vaporizes from this reservoir through the headspace (the spacer layer) to form a purple complex of tri-iodide starch in the detection reservoir of the third “acceptor layer”. Photographic images of the purple complex are analyzed using ImageJ to obtain the color intensities for the quantitation of mercury. Our device serves the needs of low- and middle-income countries with limited resources in the mining areas as simple and low-cost tool capable for mercury determination in samples with complicated matrix such as soil. The device is readily implemented to assess crisis and situation of mercury emission. Production cost of the mercury µPAD is very cost-effective (7 US$/100 devices) [33]. The reagents are common and are supplied worldwide at reasonably low in costs (potassium iodide, potassium iodate, sulfuric acid and starch). In addition to the use of the mercury standards (which is unavoidable in any mercury determination), our method does not contribute hazardous waste to the environment. 

## 2. Results and Discussion

### 2.1. Principle of Indirect Colorimetric Detection of Mercury

Mercuric ion reacts with iodide ion to form a stable complex of tetraiodomercurate (HgI_4_^2−^, K_f_ = 7.4 × 10^12^) [34]. The complex is a colorless compound which cannot be detected using a digital camera. In order to generate intense visible color, the iodometric reaction was employed. The detection principle is based on the following reactions:
**At the Donor layer**

‘sr’ reservoir:  Hg^2^^+^_(aq)_ + I^−^_(aq)_→ HgI_4_^2^^−^_(aq)_ + I^−^_(aq)_
    excess

unreacted
‘dr’ reservoir: 5I^−^_(aq)_ + IO_3_^−^_(aq)_ + 6H^+^_(aq)_→ 3I_2(g)_ + 3H_2_O_(l)_

unreacted

**At the Acceptor layer**I_2(g)_ + I^−^-starch→ I_3_^−^ -starch_(aq)_


purple complex


The first step of the reaction occurs in the donor layer ‘D’ at the sample reservoir ‘sr’ (Figure 1b) where the stable complex of HgI_4_^2^^−^ is formed. The remaining I^−^ flows to the donor reservoir ‘dr’ and is oxidized by acidic iodate to produce iodine gas (I_2(g)_). The gas diffuses across the spacer layer to react with the iodide-starch reagent producing the purple tri-iodide starch complex on the acceptor layer ‘A’. The higher the concentration of Hg(II), the less amount of iodide will remain in the donor reservoir ‘dr’, resulting in decrease of production of iodine gas and consequently decrease in the color intensity of the I_3_^−^-starch complex, as can be visually observed. The color of the complex at the acceptor zone (layer A in Figure 1) is recorded using the digital camera and the image analyzed using ImageJ software. The RGB (red, green, blue) intensity values were evaluated. It was found that the data for the green intensity scale provided the highest sensitivity of the linear calibration line (see Appendix A for the selection of the green color intensity).

### 2.2. Optimization of Physical Parameters of the µPAD

Three physical parameters were optimized using standard Hg(II) solutions in the range of 50–350 mg L^−1^ by following the protocol illustrated in Figure 2. The sensitivity, i.e., the slope of the calibration line, was the target of the optimization process.

#### 2.2.1. Diameter of Donor Reservoir

In the previous work of membraneless gas-separation µPADs reported by Phansi et al. for determination of volatile (ethanol) and non-volatile compounds (S^2^^−^ and NH_4_^+^) [32] three µPADs with donor reservoir diameters of 6, 8 and 10 mm were tested. They found that the volume of the air gap increases with an increase in the donor reservoir diameter which results in a decrease in the sensitivity of measurements. Therefore, the smallest diameter of the donor reservoir (6 mm) was selected and this diameter was also used as the size of the acceptor reservoir. When this dimension was used in this work, the color of the complex developed towards one side of the acceptor reservoir (see photographs in Table 1). As illustrated by the drawings in Table 1, for the donor reservoir with larger diameter (6 mm), it appears that iodine is generated mostly at the entrance of the donor reservoir ‘dr’. This gives rise to uneven distribution of the color and low precision of the mean of the color intensity value of the designated area of the acceptor reservoir. To solve this problem, the diameter of the donor reservoir was reduced to 3 mm and the length of the connecting channel was slightly increased from 5 to 6.5 mm. In the fabrication of the membraneless gas-separation device, the center of the 3 mm donor reservoir ‘dr’ was aligned with the center of the acceptor reservoir, as shown in the drawing in Table 1. With use of the smaller diameter of 3 mm, homogeneity of color distribution, as well as higher sensitivity of analysis, was achieved. Therefore, the donor reservoir diameter of 3 mm was selected in further optimization study.

#### 2.2.2. Effect of Reaction Time

In this work, the reaction time is defined as the time period from the addition of the iodide solution into the sample reservoir ‘sr’ (Step 5 in Figure 2) to the recording of the image of the detection area (Step 6). The reaction time was varied from 2 to 8 min, respectively. Figure 3a shows that the sensitivity significantly increased when the reaction time was increased from 2 min to 4 min. However, the sensitivity remained constant for reaction times of 6 and 8 min. This implied that the partial pressure of the iodine gas was attained within 4 min. Therefore, the minimum reaction time of 4 min was selected for further studies. However, in order to increase sample throughput, multiple analyses can be carried out since reaction time longer than 4 min does not affect the measurement.

#### 2.2.3. Effect of Spacer Thickness

Effect of the spacer thickness of the µPAD was also optimized. The spacer thickness was increased from 0.8 to 3.2 mm by increasing the number of layers of the 0.8 mm mounting tape, respectively. The result in Figure 3b shows that the sensitivity significantly decreases with increasing spacer thickness. This may be due to the increased volume of the air-gap which will require more iodine to provide the same equilibrium partial pressure and, hence, less iodine reacting with the iodide-starch reagent. This effect was also observed in the previous report [35]. Thus, we selected 0.8 mm air gap in the further studies. 

### 2.3. Optimization of Chemical Parameters of µPAD

#### 2.3.1. Iodide Concentration

The determination of Hg(II) is based on the measurement of the iodide remaining after reaction with mercury in the sample. Therefore, the initial concentration of iodide is an important parameter as it will affect the amount of iodine gas produced and, thus, the formation of the I_3_^−^-starch complex. The study was carried out using a standard solution of 150 mg L^−1^ Hg(II) with the concentrations of potassium iodide varied from 5 to 14 mmol L^−1^, respectively. As expected, increasing the initial concentration of iodide, the value of the green intensity of I_3_^−^-starch complex also increased, corresponding with the amount of remaining iodide (Figure 4b). The images of the purple I_3_^−^-starch complex are shown in Figure 4a. Measurements of the sensitivity of determination, using standard Hg(II) solutions from 50–350 mg L^−^^1^, showed increasing sensitivity for iodide concentration up to 10 mmol L^−1^. However, there is a steep decline at higher concentration of iodide (Figure 4b). Using large amount of initial iodide (>10 mmol L^−1^), the green intensity did not significantly change with the concentrations of standard Hg(II) (50–350 mg L^−1^). This showed that the amount of Hg(II) in the samples consumed only a small proportion of the initial iodide, resulting in calibration curves with low sensitivity. In fact, for loading of 8.0 µL of 10 mmol L^−1^ iodide (Step 5 in Figure 2) on to the previously applied Hg(II) standard (3.0 µL, Step 2 in Figure 2) the amount of remaining iodide is 96% for 50 mg L^−1^ Hg and 74% for 350 mg L^−1^ Hg. Thus, 10 mmol L^−1^ was selected because it provided the highest sensitivity. 

#### 2.3.2. The pH of Mercury Solution

The pH of the mercury solution was investigated to find the suitable value for the reaction of iodide to form the HgI_4_^2−^ complex. Mercuric ion is unstable at high pH where the main species include HgOH^+^, Hg(OH)_2_ and Hg(OH)_3_^−^ [36,37]. Therefore, low pH was selected to have Hg^2+^ as the main species. Optimization was carried out using standard 150 mg L^−1^ Hg in the pH range of 0.5 to 4. The results in Appendix A shows that the green intensity value increased from pH 0.5 to pH 2 and then decreased. It was found that self-oxidation of iodide was taking place at extremely low pH producing iodine gas (at sample reservoir ‘sr’) at the same as reacting with mercury. Stable condition was achieved at pH 1 and 2 (Appendix A). Kim et al. have reported that mercury in solution with +2 oxidation state has the highest stability at pH 2 and decreasing with increasing pH [37]. Thus, we selected pH 2 as the optimum pH for determination of mercury since it provides the highest green intensity with absence of self-oxidation of iodide.

### 2.4. Analytical Performance and Interference Study

#### 2.4.1. Analytical Features

Using the optimal parameters (described above) and operating procedure (Section 3.4 and Figure 2), the linear working range was 50–350 mg L^−1^ Hg, with coefficient of determination r^2^ of 0.996 (Figure 5). The limit of detection (3σ of y-intercept/slope) and limit of quantification (5σ of y-intercept/slope; [38]) were 20 mg L^−1^ and 33 mg L^−1^ Hg, respectively. Thus, the method is applicable for determination of mercury in contaminated samples such as soil including water samples collected from gold mining areas. The contaminated water samples can be directly analyzed without any sample preparation step whereas the contaminated soil samples have to be digested prior to the analysis. The mercury contents in water and digested soil samples (concentration range of a hundred mg L^−1^) are mutual with the calibration range. The repeatability of the proposed method was performed by measurement of standard Hg(II) solution of 150 mg L^−1^ Hg for twelve replicates, with a %RSD of 2.2%. The determination of one mercury sample can be carried out within 10 min; however, the number of sample throughput can be increased with simultaneous analyses of samples. 

#### 2.4.2. Interference Study

The proposed µPAD method was developed for the determination of mercury in soil and water samples from gold mining areas. Therefore, the effect of interfering species that are normally present in water and soil in such areas were investigated. According to the previous report of Malehase et al. in 2016 [39], several metal ions and anions have been found in soil and water samples in the gold mining area. These include iron (Fe), copper (Cu), chromium (Cr), cadmium (Cd), zinc (Zn), lead (Pb), silver (Ag), nitrate (NO_3_^−^), sulfate (SO_4_^−^) and chloride (Cl^−^). Therefore, these and other ions (see Table 2) were investigated for their effect on the mercury analysis. A 50 mg L^−1^ of mercury standard solution was employed as the test solution which was spiked with various concentrations of each interfering species. In this work, the tolerance limit is defined as the highest concentration of the foreign species causing a change of the green intensity less than ±1 SD (standard deviation) for the determination of 50 mg L^−1^ Hg standard solution. This SD value was 4.5% of the mean intensity. The results in Table 2 show that, among the metals ions, Fe(III) and Ag(I) show low levels of tolerance because Fe(III) acts as oxidizing agent of iodide (E^0^_Fe3+/Fe2+_ = 0.77 V, E^0^_I2/I−_ = 0.54 V) and Ag(I) can precipitate out with iodide (AgI_(s);_ K_sp_ = 8.3 × 10^−17^). As for the anions, sulfide (S^2−^) shows the lowest level of tolerance of 25 mg L^−1^ since sulfide has a high affinity for mercury to form HgS_(s)_ (K_sp_ = 2 × 10^−54^) [34]. However, the tolerance concentrations of all substances are still higher than the reported levels in water and soil samples. Therefore, the proposed method has the potential for analysis of mercury in water and soil samples without interference by possible foreign species.

### 2.5. Applications and Validation

Three soil samples (S1–S3) were obtained from artisanal small-scale gold mining (ASGM) areas on Lombok Island, Indonesia. The other soil samples (S4–S10) were collected from Bangkok City and Samut Sakorn Province, Thailand. Water samples were collected from canals in Bangkok, Thailand. Soil samples were first digested using USEPA 3050B [42] standard method and the water samples were treated as described in Section 3.2. 

Table 3 gives the amount of Hg(II) obtained from samples of the soil and water, as the concentration of the sample solution (mg L^−1^ Hg) loaded on the membraneless gas-separation µPAD (Step 2 in Figure 2). Examples of the images of the colored product that was formed in the acceptor reservoir during analysis of soil and water samples are depicted in Appendix A. Only soil samples S1–S3 were found to have Hg(II) levels. Converting the values in Table 3 to mg kg^−1^ soil sample, the amounts are 3041, 3166 and 3151 mg kg^−1^ Hg, respectively. The mercury contents in all contaminated soils are above World Health Organization (WHO) limits for agricultural soils (0.05 mg kg^−1^) [43].

Percent recoveries of spiked mercury in soil and water samples were carried out using the developed µPAD. All samples were spiked at 100 mg L^−1^ Hg, except for soil samples S1 to S3 which were spiked at 50 mg L^−1^ Hg. As shown in Table 3, the recoveries were 96.7–108.2% and 90.7–102.9% for soil and water samples, respectively. According to the AOAC guideline [44], the recovery values are in the acceptable range.

The concentrations of mercury in the soil and water samples were also analyzed using inductively coupled plasma mass spectrometry (ICP-MS). The three digested soil samples (S1–S3) were analyzed directly using ICP-MS with appropriate dilution. The other digested soil samples (S4–S10) were spiked at 2500 mg kg^−1^ Hg, whereas water samples (W1–W4) were spiked at 100 mg L^−1^ Hg. Our method provides comparable results to the reference ICP-MS method (see Figure 6) as shown by using paired *t*-test at 95% confidence level (*t*_stat_ =1.38, *t*_crit_ = 2.16). The results in Figure 6 suggest that our method is accurate and that the air-conditioning system provides good control of the temperature for gas diffusion inside µPADs.

### 2.6. Comparison of the Proposed Method with Existing Methods Including Other PADs/µPADs Methods for Determination of Mercury

Analytical features of the proposed method and previous methods are summarized in Table 4. Cold vapor-AAS technique (Technique No. 1 in Table 4) is usually the gold standard method that utilizes a very reliable equipment, the atomic absorption spectrophotometer. In this method, mercury in water or soil samples is reduced using SnCl_2_ or NaBH_4_ to form Hg^0^ [7,9,45]. However, during the procedure, the generated Hg^0^ vapor is prone to be released into the workplace which causes a health risk for the operator. Apart from the equipment-based techniques, like the cold vapor-AAS, there are alternative ways to analyze mercury without liberating Hg^0^ and without use of equipment, viz. using paper-based analytical techniques. These paper-based techniques include both PADs and µPADs which are listed in Table 4 as techniques No. 2.1–2.3 and techniques No. 3.1–3.4. The strategy of techniques No. 2.1–2.3 exploit the special features of nanoparticles (both unmodified [26,27,30] and functionalized synthesized [28,29]) to enhance the sensitivity of mercury detection. Nonetheless, the use of nanoparticles also has the problem of possible ecological issue since it is easily released into the environment [46]. Risks of nanoparticles to human health have always been an issue of public concern along with the advantages in their applications [47]. Techniques No. 3.1 to 3.3 in Table 4 show PADs/µPADs techniques that were developed without using nanoparticles. Instead, these techniques utilized a chemical reaction to form a colored compound for semi-quantitative [24] or quantitative analyses of mercury [25,31]. Although these techniques do not produce Hg^0^, the chemicals employed are not eco-friendly. 

It is also observed from Table 4 that all PADs/µPADs techniques developed in the last decade [25,26,27,28,29,30,31] were tested only in water samples (all spiked with mercury), which are not as complicated as soil. There were no alternative methods to the cold vapor-AAS for assessing soil contamination particularly for use in countries with limited resources. Our mercury µPAD was developed with the purpose of analysis of soil samples. Although our method provides less sensitivity compared to others μPADs method, the detection limit is sufficient for determination of mercury in soils from contaminated areas. The results in Figure 6 show that our µPAD is capable of analyzing real soil samples collected from artisanal small-scale gold mining areas. We also successfully validated our µPAD method for water analysis. As shown in the remark column of Table 4, our technique (techniques No. 3.4) employs very common reagents which are not hazardous. The chemicals can be easily obtained from worldwide suppliers at economic prices. The analysis time of our µPAD is comparable with most PADs/µPADs techniques [28,29,30] (10–15 min/analysis). Comparing to PAD technique No. 2.2, which is the pioneering work in utilizing AgNPs in paper-based devices for mercury [27], our method gives a much faster analysis time (10 min compared to 45 min).

## 3. Materials and Methods

### 3.1. Chemicals and Reagents

Chemicals used in this work were all analytical reagent (AR) grade. Deionized water (18.2 MΩ·cm, Thermo Scientific Easypure II system, Waltham, MA, USA), was employed for the preparation of all aqueous solutions. All of the glassware and the bottles were cleaned, rinsed with deionized water, soaked overnight in 10% (*v*/*v*) nitric acid, then rinsed with deionized water. 

A standard stock solution of Hg(II) (10.0 g L^−1^ Hg) was prepared by dissolving an accurate weight of 1.36 g of mercuric chloride powder (Ajax Finechem., Australia) in 0.01 mol L^−1^ of nitric acid (RCI Lab Scan, Thailand) in a 100 mL volumetric flask. Working standard solutions of mercury were prepared by subsequent dilution of this stock solution with 0.01 mol L^−1^ nitric acid. A stock solution of 0.5 mol L^−1^ potassium iodate was prepared by dissolving 1.07 g of potassium iodate powder (Ajax Finechem.) in 100 mL of deionized water. This solution was used for preparing the oxidizing agent for iodometric reaction (at donor reservoir ‘dr’), which is a mixture of 0.2 mol L^−1^ potassium iodate and 0.2 mol L^−1^ sulphuric acid. The solution was freshly prepared daily by mixing 4 mL of 0.5 mol L^−1^ potassium iodate and 4 mL of 0.5 mol L^−1^ sulphuric acid (RCI Lab Scan, Thailand) and then made up to 10 mL with deionized water. Potassium iodide solution is used for forming the HgI_4_^2^^−^ complex at the sample reservoir (‘sr’ in Figure 2). A stock solution of 0.1 mol L^−1^ potassium iodide was firstly prepared by dissolving 1.66 g of potassium iodide crystal (Merck, Germany) in 100.0 mL of deionized water. The solution of 10 mmol L^−1^ potassium iodide was prepared by diluting this stock solution 10-fold with deionized water. The solution of 1% (*w*/*v*) of starch in 0.1 mol L^−1^ potassium iodide was used as the color forming reagent at the acceptor reservoir (‘ar’ in Figure 2). The solution was freshly prepared daily by dissolving 0.05 g of starch powder (BDH, U.K.) in 5 mL boiling deionized water and then letting the solution cool to room temperature before adding 0.083 g potassium iodide crystal.

### 3.2. Preparation of Samples

The water samples were filtered with Whatman filter paper No. 42 (Whatman International, Ltd., Maidstone, UK) followed by acidification with nitric acid to pH 2. For soil samples, the wet soil/sediments were placed in a petri dish and air dried for 48 h. The dried soil samples were ground with a mortar and pestle and then heated in an oven at 120° C for 2 h. After cooling to room temperature, the samples were stored in a desiccator until analyzed. The soil samples were acid digested prior to analysis. The USEPA 3050B protocol of the standard acid method for determination of heavy metals in soil samples [42] was adopted for this work. Previous comparison study [49] have shown that there is an acceptable correlation (r = 0.99) between the USEPA 3050B protocol (open system digestion) and the USEPA 3051A protocol (close system digestion by microwave) for determination of mercury in 10 classes of soils. Briefly, 2 g of soil was placed in a Teflon beaker; then, 5 mL of concentrated HNO_3_ was added and the beaker covered with a watch glass. The solution was then heated on a hotplate at 95 ± 5 °C for 15 min and cooled to room temperature. Another 5 mL of concentrated HNO_3_ was added and heated again at 95 ± 5 °C for 2 h. After cooling 2 mL of deionized water and 3 mL of 30% hydrogen peroxide (Sigma-Aldrich, Germany) were added, followed by the heating step for a further 2 h. The last step of digestion was the addition of 10 mL of hydrochloric acid (RCI Lab Scan, Thailand) with heating for 15 minutes at 95 ± 5 °C. The digested solution was filtered with 0.45 µm of cellulose acetate membrane and adjusted to pH 2 using a few drops of 50% (*w*/*v*) sodium hydroxide. The filtered solution was made up in a 50 mL volumetric flask with deionized water. This solution was used for the determination of mercury using the membraneless gas-separation µPAD. Appropriate dilution of the filtrate with 2% (*v*/*v*) sub-boiled nitric acid was performed prior to determination of Hg in the samples by inductively coupled plasma mass spectrometry (ICP-MS 7900, Agilent Technologies, Santa Clara, CA, USA) as the comparison method. 

### 3.3. Fabrication of µPAD 

The membraneless gas-separation µPAD comprises 3 layers: a donor layer, a spacer layer and an acceptor layer. Two patterns of the hydrophobic barriers were employed, as shown in Figure 1a: a single circular pattern for the donor layer and a dumbbell-shaped pattern with 2 different sizes of the reservoirs for the donor layer. The sample reservoir (‘sr’) and the acceptor reservoir (a’r)’ were of the same dimension of 6 mm i.d., whereas the second donor reservoir (‘dr’) has 3 mm i.d. The spacer layer was a mounting tape with 0.8 mm thickness (Scotch™, St. Paul, MN, USA) which has a 7 mm circular disc cut out. The donor layer, spacer layer and acceptor layer were assembled together to produce the membraneless gas-separation µPAD as shown in Figure 1b. Figure 1c shows the assembled µPAD from the acceptor and donor side, respectively. The printed circular hydrophobic barriers are screen-printed on a Whatman No. 4 filter paper (Maidstone, UK), following the method of Sitanurak et al. [33], to define the circular reservoirs for the donor and acceptor layers, respectively. The patterned paper was kept overnight at the room temperature for complete curing of the resin prior to cutting to give separate devices. One screen-printed paper provides 77 pads. These µPAD devices are stable for at least 2 years.

### 3.4. Operating Procedure 

The operating procedure for using the membraneless gas-separation µPAD to measure the amount of Hg(II) is shown in Figure 2. The analysis is carried out at room temperature in an air-conditioned room (26–27 °C). In the first step (Step 1), 1.0 µL of the oxidizing agent (0.2 mol L^−1^ KIO_3_ in 0.2 mol L^−1^ H_2_SO_4_) is loaded on the donor reservoir ‘dr’ using an autopipette (Rainin Instrument, Mettler Toledo, Switzerland), followed by (Step 2), the addition of 3.0 µL Hg(II) standard/sample on to the sample reservoir ‘sr’ and then waiting ca. 5 min for the pad to dry. Next, the device is turned over and (Step 3) 2.0 µL of the color developing reagent (1% (*w*/*v*) starch in 0.1 mol L^−1^ KI) is loaded on the acceptor reservoir ‘ar’. The acceptor reservoir ‘ar’ and donor reservoir ‘dr’ are then immediately covered with transparent adhesive tape (Scotch™) (Step 4) to prevent loss of the reagents from the device. Next, (Step 5), 8.0 µL of 10 mmol L^−1^ KI is loaded on the sample reservoir ‘sr’. The iodide solution reacts with Hg(II) deposited earlier (in Step 2) forming the colorless HgI_4_^2^^−^ complex at the ‘sr’ reservoir. The unreacted iodide diffuses into reservoir ‘dr’, where it is converted to volatile iodine (I_2(g)_) by iodate ion deposited earlier (in Step 1). The iodine gas diffuses across the spacer to react with the starch-KI reagent on the acceptor side (Step 3) to generate the purple color of triiodide-starch complex. In the final step (Step 6) the acceptor layer is placed face up inside an in-house light-box (fitted with a JMF fs-wh-1 watt LED light tube). The image of the purple color complex is recorded at 4 min after the addition of KI (Step 5 in Figure 2) using a digital camera (IXUS 125 HS, Canon, Japan). The color intensity of the image is analyzed using ImageJ software (version v1.35e). The green color scale is used to construct the calibration line for concentrations of Hg(II) in the standard solutions. Note: The volume of sample of 3.0 µL was chosen to provide the maximum loading of the sample into the sample reservoir ‘sr’ without the overflow of sample into the donor reservoir ‘dr’. 

## 4. Conclusions

A simple, cost-effective, equipment-free and environmentally friendly µPAD method was developed for determination of Hg(II) in contaminated soil and water samples by using a membraneless gas-separation µPAD. Mercury in the sample is determined by an indirect colorimetric procedure. The detection principle is based on quantitating the color image of I_3_^−^-starch complex. In the donor layer of the membraneless gas-separation µPAD, the pre-deposited sample containing mercuric ions reacts with an added solution containing a fixed amount of excess iodide to form the colorless complex of HgI_4_^2^^−^. The remaining iodide flows to a connected adjacent area, where it is oxidized by acidic iodate to produce volatile iodine via the Dushman reaction [50]. The volatile iodine moves across an air spacer region to react with pre-deposited iodide-starch to form the purple I_3_^−^-starch complex at the acceptor layer. This method is applicable for measurement of high concentrations of mercury, especially in contaminated soil and water in artisanal small-scale gold mining area. 

The method fits the purpose for waste management of contaminated sites in many developing countries where artisanal small-scale gold mining is still an important primary economic sector (e.g., Asia, Africa and South America) [2]. Our developed µPAD for mercury detection has several advantages. The fabrication of µPAD by screen-printing technique is simple and does not require sophisticated skill. Hence, this mercury µPAD can be produced anywhere in the world where filter paper, double-sided mounting tape, screen-printing tools, t-shirt ink (as hydrophobic barrier) are available. The production cost is approximately 7 US$/100 devices [33] which is suitable for developing countries. Moreover, our method is in compliance with “Green Analytical Chemistry”, since there is reduction of waste generation (e.g., no hazardous synthesis), with low human toxicity and eco-toxicity.

## Figures and Tables

**Figure 1 molecules-26-02004-f001:**
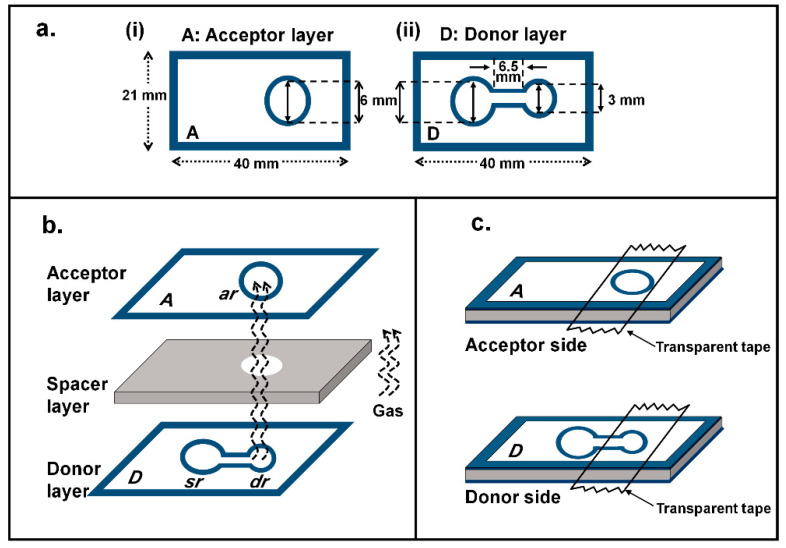
(**a**) The µPAD pattern: (i) acceptor layer A with circular-shaped barrier and (ii) donor layer D with dumbbell-shaped barrier. (**b**) The three layers of the membraneless gas-separation µPAD, showing alignment of the donor layer, the spacer layer with circular hole and the acceptor layer. (**c**) 3D-view of assembled device from both the acceptor and donor sides, with position of the transparent tapes.

**Figure 2 molecules-26-02004-f002:**
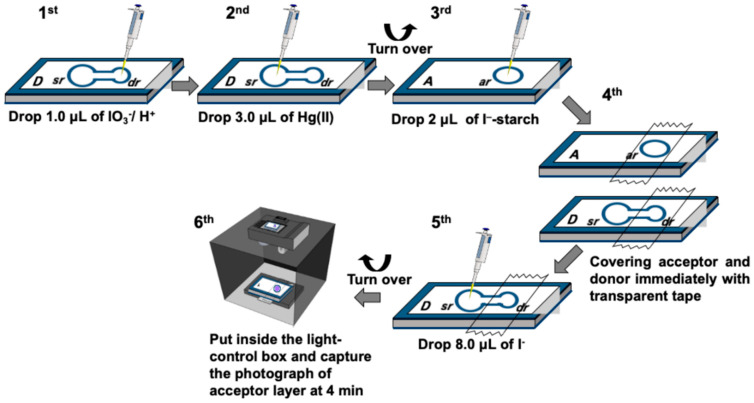
Illustration of the operating steps for the determination of mercury using the membraneless gas-separation µPAD.

**Figure 3 molecules-26-02004-f003:**
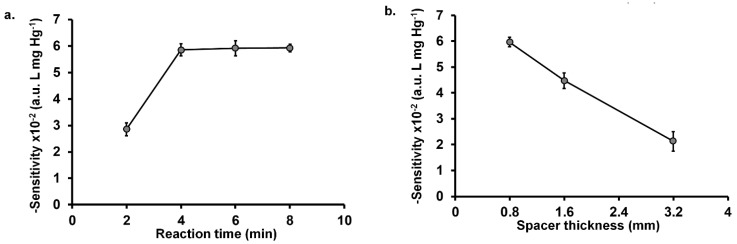
Effect of physical parameters on analysis of Hg(II): (**a**) reaction time and (**b**) spacer thickness. Experimental conditions: 0.2 mol L^−1^ KIO_3_ in 0.2 mol L^−1^ H_2_SO_4_, 10 mmol L^−1^ KI and 1% (*w*/*v*) of starch in 0.1 mmol L^−1^ KI. For the spacer thickness study, the reaction time is 4 min.

**Figure 4 molecules-26-02004-f004:**
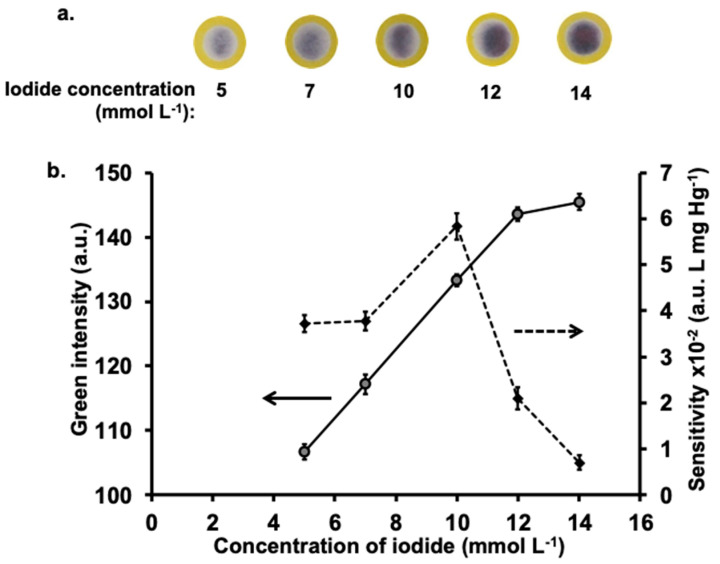
Effect of iodide concentration on analysis of Hg(II): (**a**) images of purple iodine-starch complex for various concentrations of iodide analysis using a standard solution of 150 mg L^−1^ Hg and (**b**) plots of effect of iodide concentrations on the green intensity values (left ordinate) using a standard solution of 150 mg L^−1^ Hg and sensitivity of Hg(II) analysis (right ordinate). Experimental conditions: 0.2 mol L^−1^ KIO_3_ in 0.2 mol L^−1^ H_2_SO_4_, 10 mmol L^−1^ KI, 1% (*w*/*v*) of starch in 0.1 mmol L^−1^ KI and reaction time of 4 min.

**Figure 5 molecules-26-02004-f005:**
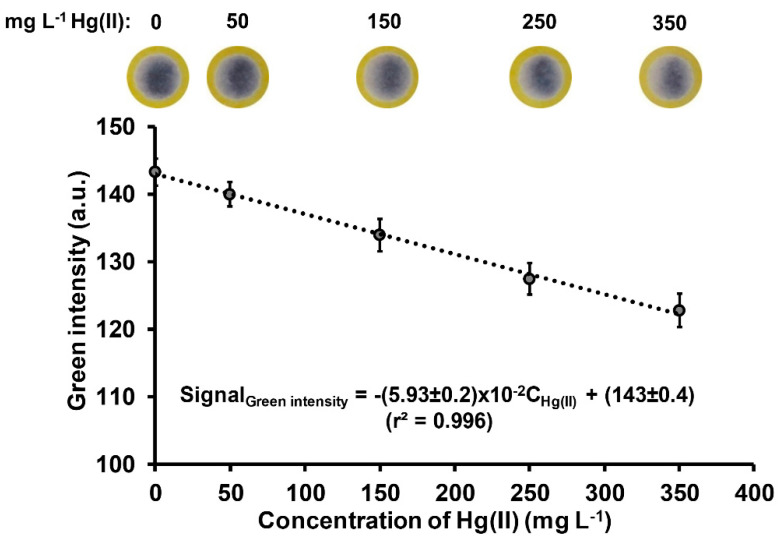
Calibration line using the membraneless gas-separation µPAD for the determination of mercury and the corresponding image of the purple iodine-starch complex.

**Figure 6 molecules-26-02004-f006:**
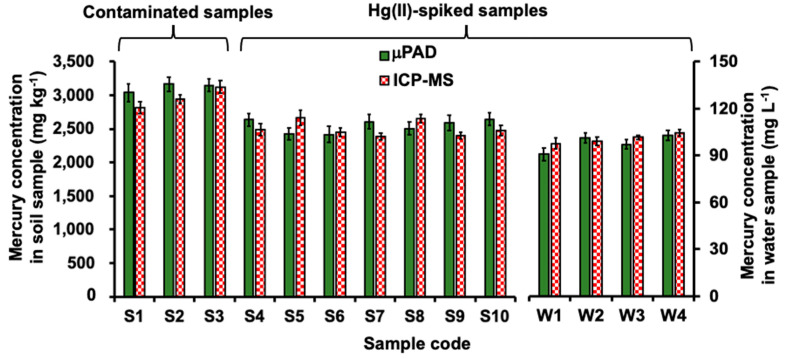
Bar plots of the concentrations of mercury in digested soil and water samples as determined using the membraneless gas-separation µPAD and the reference ICP-MS method. The digested soil samples, S1–S3, were analyzed directly using ICP-MS with appropriate dilution. The other digested soil samples, S4–S10, were spiked at 2500 mg kg^−1^ Hg. The water samples, W1–W4, were spiked at 100 mg L^−1^ Hg.

**Table 1 molecules-26-02004-t001:** Effect of size of diameter of donor reservoir on homogeneity of color distribution at the acceptor reservoir and on the sensitivity of analysis.

Diameter of Donor Reservoir	Working Range(mg L^−1^ Hg)	Linear Equation	Schematic Diagram of the Experimental Study	Image of the Acceptor Reservoir (6 mm) for 150 mg L^−1^ Hg
6 mm	50–350	Intensity- = (4.0 ± 0.5) × 10^−2^ C_Hg(II)_ + (123.7 ± 1.1)	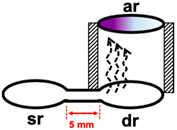	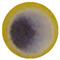
3 mm	50–350	Intensity- = (5.5 ± 0.5) × 10^−2^ C_Hg(II)_ + (127.9 ± 1.1)	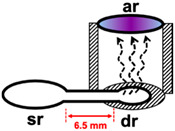	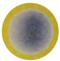

**Table 2 molecules-26-02004-t002:** The tolerance limit of the µPAD for analysis of Hg(II) and comparison with the reported levels of foreign species in some samples (drinking water, river water and soil).

Foreign Species	Reported Level		Tolerance Limit (mg L^−1^)
	River Water (mg L^−1^) ^a^	Soil (mg kg^−1^) ^b^	
Copper (II)	0.006–10	5–70	1000 ^c^
Lead (II)	0.003–0.3	10–67	1000 ^c^
Cadmium (II)	0.01–0.04	6.4–11.7	1000 ^c^
Iron (II)	0.03–0.05	0.5–10	1000 ^c^
Nitrate (NO_3_^−^)	5–50	8–119	1000 ^c^
Sulfate (SO_4_^2−^)	–	29–130	1000 ^c^
Cyanide (CN^−^)	≤0.2	11–44	750
Chromium (III)	0.05–0.2	2–60	500
Zinc (II)	0.05–0.1	8.9–65.7	500
Nickel (II)	0.03–10	3–100	500
Iron (III)	≤7	20–30	250
Silver (I)	0.3–1	0.2–0.3	250
Sulfide (S^2−^)	Up to 0.05	Up to 11.7	25

^a^ Reported by World Health Organization, 2011. [40]; ^b^ Reported by Fashola et al. [41]; ^c^ Maximum tested concentration.

**Table 3 molecules-26-02004-t003:** Percentage recovery of mercury in soil and water samples using the µPAD.

Sample	Mercury Concentration (mg L^−1^ Hg)			% Recovery
	Present ^a^	Added	Found ^b^	
Soil Sample				
S1	121.7 ± 5.3	50	174.4 ± 4.5	105.5
S2	126.7 ± 4.3	50	175.2 ± 3.4	97.0
S3	126.1 ± 3.7	50	180.2 ± 4.2	108.2
S4	n.d.	100	105.4 ± 3.6	105.4
S5	n.d.	100	96.9 ± 3.6	96.9
S6	n.d.	100	96.7 ± 4.9	96.7
S7	n.d.	100	104.1 ± 4.3	104.1
S8	n.d.	100	100.3 ± 3.8	100.3
S9	n.d.	100	103.5 ± 4.6	103.5
S10	n.d.	100	105.7 ± 3.7	105.7
Water Sample				
W1	n.d.	100	90.7 ± 3.9	90.7
W2	n.d.	100	101.2 ± 3.5	101.2
W3	n.d.	100	97.2 ± 2.9	97.2
W4	n.d.	100	102.9 ± 3.2	102.9

^a^ Mean concentration ± SD, *n* = 3. ^b^ Concentration of sample after spiking with standard solution; n.d.: Not detected.

**Table 4 molecules-26-02004-t004:** Comparison of the analytical features of various techniques for the determination of mercury.

Technique Class/Reagent	Test Samples	Working Range/LOD	Analysis Time (Excluding Sample Preparation)	Classified as “Equipment-Free” Method	Remark
**1. Cold vapor-AAS/**SnCl_2_, NaBH_4_, KMnO_4_(for hydride generation)	Water [45]	0.04–2.4 μg L^−1^ Hg/0.02 μg L^−1^ Hg	NR ^a^	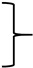	No	All reagents are commercially available.Hg vapor is generated.
Soil [9]	5–40 μg L^−1^ Hg/0.01 μg L^−1^ Hg	NR ^a^
Soil and sediment [7]	1–30 μg L^−1^ Hg/0.08 mg kg^−1^	NR ^a^
**2. Colorimetric** **μPADs or PADs with NPs/**				
2.1 Functionalized AuNPs, TMB, H_2_O_2_	Tap water (spiked) [29]	0.2–2000 ng Hg/30 μg L^−1^ Hg	10 min	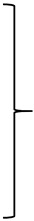	Yes	The factionalized AuNPs are not yet commercialized.
2.2 Synthesized-AgNPs	Drinking water (spiked), tap water (spiked) [30]	0.01–10 mg L^−1^ Hg/0.003 mg L^−1^ Hg	12 min	AgNPs may be synthesized or purchased.
	Drinking water (spiked), tap water (spiked) [26]	0.05–7 mg L^−1^ Hg/0.001 mg L^−1^ Hg	NR
	Drinking water (spiked), tap water (spiked) [27]	5–75 mg L^−1^ Hg/0.12 mg L^−1^ Hg	45 min
2.3 Synthesized-CcNPs	Industrial water (spiked) [28]	0.5–20 mg L^−1^ Hg/0.17 mg L^−1^ Hg	15 min	CcNPs are not yet commercialized.
**3. Colorimetric** **μ** **PADs or PADs without NPs/**				
3.1 Dithizone in CCl_4_	Synthetic water, whitening cream extract [31]	1–30 mg L^−1^ Hg/0.93 mg L^−1^ Hg	≥10 min	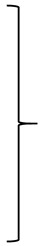	Yes	Not eco-friendly reagent/solvent.
3.2 Resorufin thiono-carbonate in CH_2_Cl_2_	Simulated wastewater [25]	2–10 mg L^−1^ Hg/1.18 mg L^−1^ Hg	NR	Not eco-friendly reagent/solvent.
3.3 Rhodamine derivatives in CH_2_Cl_2_	NONE [24]	20, 50, 100, 200, 300 mg L^−1^ Hg (no calibration plot)/NR	≥15 min	Need synthesized chemicals.Not eco-friendly solvent.
3.4 ***This work***/KI, KIO_3_, H_2_SO_4_, starch	Water (spiked)Soil (from gold mining)	50–300 mg L^−1^ Hg/20 mg L^−1^ Hg	10 min	All reagents are common and are all commercially available.No serious toxicity from skin exposure (KI, KIO_3_, starch) except 0.2 mol L^−1^ H_2_SO_4_.

NR: Not reported; TMB: 3,3′,5,5′-tetramethylbenzidine; CcNPS: Curcumin nanoparticles. ^a^ Usual analysis time for cold vapor-AAS is 3 min with use of flow injection system [48].

## Data Availability

The data presented in this study, are available on request from the corresponding authors.

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
