# Peer review of "Simple and Equipment-Free Paper-Based Device for Determination of Mercury in Contaminated Soil"

_molecules, 2021, doi:10.3390/molecules26072004_

Round 1
Reviewer 1 Report
This worked introduces a paper-based method for the determination of mercury in soil samples. The colorimetric sensing of the mercury relies on the color change of iodine. The color intensity was analyzed using ImageJ and calibration curve ranged between 50-350 mg/L. My comments are as below: 1- The acceptable level (range) of the mercury in soil should be included in the Abstract. 2- Equations in page 3 & 4 should be revised. Also, parentheses in whole manuscript should be corrected. 3- English level of the manuscript should be thoroughly revised. 4- In order to have a better comparison between proposed method and ICP-MS, it is recommended to provide figure 6 colorful. 5- Did the Author optimize the dispensing volume of the sample? 6- Which version of ImageJ was utilized? This should be mentioned in the main text. 7- How is the stability of the fabricated device? 8- It would be better to include more images of real devices subjected to different concentrations of mercury samples.Author Response
Please see the attachment.

Reviewer 2 Report
This research presents a microfluidic paper- based analytical device as simple and low-cost tool capable for mercury determination in water and soil.
Strength
1. the method is a simple and low-cost tool with the potential for analysis of mercury in water and soil samples without interference by possible foreign species.
2. 4-10 min reaction time
3. a good quality reagent development process for paper- based analytical device research
Weakness
1. Why only use green intensity? Why not use R+G+B or grayscale? Is the lighting system of the device using LED or a mercury lamp? Does the intensity of the illumination light affect the sensitivity of the results?
2. What is the temperature of chemical reaction in this research? Need heating or cooling? Dose the ambient temperature affect accuracy of the test? Does the temperature affects measurement results?
3. Colormetirc area in the strip of each results is not uniform. How much percentage of the reacting area is the effective sampling area? Does the yellow hydrophobic material penetrate into the sampling area and cause errors? (in Fig. 5)
Round 2
Reviewer 1 Report
The questions were answered properly and I recommend it for publication now.